# Experimental Study on the Lubrication and Cooling Effect of Graphene in Base Oil for Si_3_N_4_/Si_3_N_4_ Sliding Pairs

**DOI:** 10.3390/mi11020160

**Published:** 2020-02-03

**Authors:** Lixiu Zhang, Xiaoyi Wei, Junhai Wang, Yuhou Wu, Dong An, Dongyang Xi

**Affiliations:** 1School of Mechanical Engineering, Shenyang Jianzhu University, Shenyang 110168, China; zhanglixiu@sjzu.edu.cn (L.Z.); jhwang@sjzu.edu.cn (J.W.); wuyh@sjzu.edu.cn (Y.W.); andong@sjzu.edu.cn (D.A.); 2Test and Analysis Center, Shenyang Jianzhu University, Shenyang 110168, China; 3National-Local Joint Engineering Laboratory of NC Machining Equipment and Technology of High-Grade Stone, Shenyang 110168, China; 4School of Material Science and Engineering, Shenyang Jianzhu University, Shenyang 110168, China; xidy12@mails.jlu.edu.cn

**Keywords:** Si_3_N_4_, graphene, lubrication, friction, temperature rise

## Abstract

Recently, the engineering structural ceramics as friction and wear components in manufacturing technology and devices have attracted much attention due to their high strength and corrosion resistance. In this study, the tribological properties of Si_3_N_4_/Si_3_N_4_ sliding pairs were investigated by adding few-layer graphene to base lubricating oil on the lubrication and cooling under different experimental conditions. Test results showed that lubrication and cooling performance was obviously improved with the addition of graphene at high rotational speeds and low loads. For oil containing 0.1 wt% graphene at a rotational speed of 3000 r·min^−1^ and 40 N loads, the average friction coefficient was reduced by 76.33%. The cooling effect on Si_3_N_4_/Si_3_N_4_ sliding pairs, however, was optimal at low rotational speeds and high loads. For oil containing 0.05 wt% graphene at a lower rotational speed of 500 r·min^−1^ and a higher load of 140 N, the temperature rise was reduced by 19.76%. In addition, the wear mark depth would decrease when adding appropriate graphene. The mechanism behind the reduction in friction and anti-wear properties was related to the formation of a lubricating protective film.

## 1. Introduction

With the development of advanced manufacturing technology, the devices are often operated under conditions such as high temperature, high pressure, or less lubrication etc. However, traditional metal materials and metal tribo-pairs were not available due to its vulnerability to rupture and corrosion damage. Engineering ceramic tribo-pairs, for example Si_3_N_4_/Si_3_N_4_, Al_2_O_3_/Al_2_O_3_, etc., have excellent properties such as low density, significant thermal stability, and high hardness [1]. These properties are suitable for a wide variety of tribological applications [2,3]. Therefore, the research and practical application of Si_3_N_4_/Si_3_N_4_ as friction and wear components has become one of the hot spots of devices and material science.

Lubrication is an important measure to reduce wear, save energy, and improve industrial efficiency and reliability. In addition, lubricating oil additives are also important for improving the performance of lubricating oil. Nanomaterials as a kind of lubricant additive play a good role in anti-wear and anti-friction. Among them, as the component part of graphite used as the traditional solid lubricating material, graphene has attracted much attention in the tribological field due to its unique friction and wear properties [4,5]. Graphene possesses an extremely thin laminated structure, high load-bearing capacity, high chemical stability, and low surface energy, and thus, it can offer lower shear stress and prevent direct contact between metal interfaces [6]. Liu et al. [7] prepared novel composite coatings of diamond-like carbon/ionic liquid/graphene. They found that 0.075 mg·mL^−1^ graphene in the composite coatings exhibited the lowest friction coefficient, and the highest bearing capacity in a simulated space environment. Huang et al. [8] investigated the tribological behavior of the graphite nanosheets as an additive in paraffin oil. Their result showed that the load-carrying capacity and anti-wear ability of the lubricating oil were improved. Lin et al. [9], likewise, carried out a new kind of lubricating oil containing modified graphene platelets. The results indicated that it could clearly improve the wear resistance and load-carrying capacity of the machine.

Although many studies have shown that graphene as a lubricant additive can improve the tribological properties of sliding trio-pairs, most of the studies have focused on steel/steel tribo-pairs. In our previous study, we investigated the tribological behaviors of Si_3_N_4_/GCr15 sliding pairs lubricated with graphene oxide [10]. However, there is less attention on the lubrication of Si_3_N_4_/Si_3_N_4_ tribo-pairs, as the structure and performance characteristics of engineering ceramic device material are very different from those of metal materials. In addition, there are also few reports on the cooling effect of graphene as a lubricant additive. The purpose of this work is to explore the lubrication and cooling effect for Si_3_N_4_/Si_3_N_4_ tribo-pairs by adding graphene to base lubricating oil. Factors influencing friction coefficient include rotational speed, load, and graphene concentration. We have investigated the tribological behavior of Si_3_N_4_/Si_3_N_4_ sliding pairs under three different experimental conditions by using a Rtec MFT 5000 Tribometer with the ball-on-disk mode. Finally, the effect on lubrication and cooling and the lubrication mechanism is discussed. 

## 2. Materials and Methods 

### 2.1. Materials

Few-layer graphene (FLG) was obtained from Detong Nanotechnology Co. Ltd. (Qingdao, China). Mobile DTE oil light (Mdol) were selected as the base lubricating oil and purchased from Huijie Development Co. Ltd. (Changsha, China). FLG was obtained by physical methods and used directly without further purification. The kinematic viscosity of Mdol was 5.34 mm^2^·s^−1^, when the temperature was 100 °C and 29.77 mm^2^·s^−1^ with a temperature of 40 °C. Mdol containing different concentrations of FLG (0.025, 0.05, 0.075, and 0.1 wt%) was prepared and followed by ultrasonication for about 30 min to make sure that FLG was evenly dispersed in the base lubricating oil.

### 2.2. Tribological Test

Si_3_N_4_ ceramic balls and disks were purchased from Zhihai Bearing Co., Ltd. (Shanghai, China). The ball was a commercial product with a diameter of 9.525 mm and a surface roughness of no more than 140 nm. The thickness of the disk was 5 mm and its surface roughness did not exceed 250 nm. The tribological behaviors of Si_3_N_4_/Si_3_N_4_ tribo-pairs lubricated without and with FLG were examined using a Rtec MFT5000 Tribometer (Rtec, San Jose, CA, USA) with the ball-on-disk mode, with the cylindrical upper tribo-pairs as a Si_3_N_4_ ball and the lower tribo-pairs as a Si_3_N_4_ disk. The temperature of the Si_3_N_4_/Si_3_N_4_ sliding pairs was recorded by a temperature-rise detection device. The temperature-rise detection device included a data acquisition device, a standard rod, and the spindle error analyzer software. The temperature sensor used in data acquisition was a magnetic thermistor attached to the cylindrical pin above the Si_3_N_4_ ceramic ball. It recorded temperature changes in real time during the experiment.

In a typical test, Mdol containing 0.025, 0.05, 0.075, and 0.1 wt% FLG were prepared and ultrasonic for about 30 min to ensure uniform dispersion of the graphene in the base lubricating oil. In order to avoid interference from other factors, the graphene additive lubricating oil did not use a surfactant. Since factors influencing the tribological properties was carried out, including FLG concentration (0.025, 0.05, 0.075, and 0.1 wt%), load (40, 80, 140 N), and rotational speed (500, 3000 r·min^−1^). All tribological tests were repeated at least three times. The friction coefficient was automatically recorded by the experimental device and the real-time temperature was recorded by the temperature-rise detection system.

### 2.3. Characterization

The structure of FLG was imaged with Raman spectroscope (HR800, Horiba, Paris, France) with a confocal Raman microscope mode and a laser wavelength of 532 nm. Nanoparticle analyzer is an instrument that uses a physical method to test the size and distribution of particles. The particle diameter distribution of FLG was measured by the NanoPlus-3 nanoparticle analyzer (Micromeritics, New York, NY, USA). The wear mark depth of the Si_3_N_4_ disk was characterized by the OLS4100 3D laser measuring microscope (Olympus, Tokyo, Japan). S-4800 scanning electron microscope (SEM, Hitachi, Tokyo, Japan) equipped with an energy-dispersive X-ray spectroscope (EDS, Hitachi, Tokyo, Japan) and Raman spectroscope (Horiba, Paris, France) were used to observe the wear mark of the Si_3_N_4_ disk. 

## 3. Results and Discussion

### 3.1. Materials Characterization

Figure 1 illustrates the characterization of FLG. Figure 1a is the SEM image of FLG, showing that FLG retains its original laminated structure, which is transparent with folding at the edges, suggesting very few layers. Figure 1b portrays the Raman spectrum of FLG. Three typical features of FLG, the D-band (1342 cm^−1^), G-band (1575 cm^−1^), and 2D-band (2701 cm^−1^), are observed. As shown in Figure 1b, the peak shape of the 2D-peak is widest, the intensity of the G-peak is very strong, and the intensity ratio of the G to 2D band (*I*_G_/*I*_2D_) is more than 1, demonstrating that the graphene samples exhibit a few-layered structure [11,12], as is proved in SEM (Figure 1a). However, the presence of the D-band (1342 cm^−1^) illustrates the occurrence of disorder and defects in the graphene samples [13,14]. 

### 3.2. Tribological Properties

The average friction coefficient (COF) values of the Si_3_N_4_/Si_3_N_4_ sliding pairs lubricated by Mdol with and without FLG at different experimental conditions are shown in Figure 2. Figure 2a shows the friction coefficient curves of the Si_3_N_4_/Si_3_N_4_ lubricated by Mdol with 0.1 wt% FLG at 500 r·min^−1^ speed under 40 N loads. When FLG is added to the base oil, the friction coefficient of the Si_3_N_4_/Si_3_N_4_ sliding pairs is drastically reduced. Figure 2b shows the COF of the Si_3_N_4_/Si_3_N_4_ sliding pairs lubricated with and without different contents of FLG in three different working conditions. Figure 2b clearly shows that different amounts of graphene added to the lubricating oil have less influence on the lubrication effect at low speeds (such as 500 r·min^−1^), independent of load. Moreover, the lubricating effect is better at high speed and low load. At low speed and high load, the excessive load causes the protective film to rupture. Higher rotational speeds produce a lower COF. When the FLG content is 0.05 wt%, the COF is reduced the least, since the FLG is quickly removed from the wear track due to centrifugal force. As FLG content in base oil is increased, although the FLG is removed by centrifugal force at the same rate, the additional FLG on the wear track improves the lubricating effect depicted by a low friction coefficient. When FLG content is 0.1 wt%, the COF is reduced the most, decreasing by 76.33% (from 0.169 to 0.04).

The wear mark depth (WMD) profile curves of the Si_3_N_4_ disk lubricated by different FLG concentrations under a load of 40 N, at a rotational speed of 3000 r·min^−1^ are shown in Figure 3. It can be seen that the FLG concentration has a certain influence on the WMD of the Si_3_N_4_ disk. The WMD can be reduced when the Si_3_N_4_ disk is lubricated by a small amount of FLG. However, when the FLG concentration is too much, it will lead to aggregation and to the increased wear.

Figure 4 shows SEM images of wear scars on the Si_3_N_4_ disks lubricated by pure Mdol (Figure 4a) and Mdol containing 0.1 wt% FLG (Figure 4b). Many pores and deep scratches can be observed on the rubbing surface lubricated by pure Mdol, and EDS analysis reveals a surface carbon content of only 10.3 wt%. On the contrary, the tribo-surface lubricated by Mdol containing 0.1 wt% FLG becomes significantly smoother with less scratches and fewer pores compared to that lubricated with pure Mdol. In addition, carbon content has increased to more than 25%. This is because FLG enters the interface of tribo-pairs and forms a lubricating protective film, preventing direct contact between the sliding pairs [15].

### 3.3. Cooling Properties

Friction causes heat and temperature fluctuations during friction and wear testing. Temperature change with time for the Si_3_N_4_/Si_3_N_4_ sliding pairs lubricated by Mdol with various contents of FLG under different conditions are shown in Figure 5.

By testing different contents of FLG to the base oil, we observe that a low content of FLG is better in deterring temperature rise at low-speed (500 r·min^−1^ rotating speed). First of all, this can be attributed to the excellent physical properties of FLG (high thermal conductivity and large specific surface area) [16]. When a small amount of FLG is added to Mdol, the overall thermal conductivity of the mixed liquid is increased [17]. During the sliding process, the COF was small, and the lubricant oil flow will take away some of the heat, furthermore, FLG particles distributed in the Mdol will promote heat transfer [18]. Therefore, the heat transfer performance is improved, resulting in a smaller temperature rise. When FLG content increases, a lot of graphene will agglomerate, leading to slower lubricant flow and reduced heat transfer. Secondly, for low rotational speeds, at the same content of FLG, the temperature rise at high loads is higher. This is because the friction coefficient is higher at heavy loads, resulting in more heat generation and therefore a rise in temperature rise. Interestingly, lubricating oils with different graphene concentrations have less influence on the temperature rise at high speed and low load, which is because the centrifugal force has a large influence at high rotation speed, causing some graphene to be scooped out of the wear track. In addition, the friction generates more heat at high speed, which cannot be offset by the cooling effect of a small amount of graphene.

Since Raman spectroscopy is a superior, sensitive, and widely used non-destructive characterization technique and is also an effective tool for assessing the quality of carbon materials. In this work, we use this technique to analyze the rubbing surfaces. Figure 6 shows the Raman spectra of the wear surfaces of the Si_3_N_4_ disks lubricated by Mdol containing different contents of FLG, as well as that of original FLG powder and in the case of dry friction. In contrast to dry friction (Figure 6a), it can be inferred that the band at 2194 cm^−1^ is the typical signal of Mdol (Figure 6b). When Si_3_N_4_ disks are lubricated by Mdol containing 0.1 wt% FLG (Figure 6c), Raman spectra of the wear surfaces not only show the lubricating oil signal, but also the FLG signal at the bands 1343 and 1595 cm^−1^. This indicates that FLG is adhered to the wear surfaces and forms a lubricating protective film. Compared to the Raman signal of FLG powder (Figure 6e), for all FLG (Figure 6d) and Mdol with 0.1% FLG, the intensity ratios of the D and G bands (*I*_D_/*I*_G_) are increased, which illustrates that the adhered graphene has become severely disordered [19], and as a result, the peak shape of 2D becomes weak and wide [20]. Raman spectroscopy was carried out to further confirm that the improved tribological behavior of Si_3_N_4_/Si_3_N_4_ sliding pairs is indeed attributable to the presence of FLG on the rubbing surface. These results validate our assumption that FLG is indeed present and has formed a lubricating protective film on the wear surface.

Based on the above analysis, a schematic illustration of the tribological mechanism using FLG as a lubricant additive is shown in Figure 7. A schematic diagram of the friction experiment is shown in Figure 7a. As shown in Figure 7b, direct contact between the Si_3_N_4_ ceramic ball and disk results in high Hertz contact stress. Therefore, the oil film is quickly broken, and then, the contact surface of Si_3_N_4_/Si_3_N_4_ sliding pairs is seriously destroyed. When FLG is added to the base oil, it easily enters the rubbing interface and is sheared owing to its two-dimensional structure [21], as shown in Figure 7c. The addition of FLG into Mdol prevents direct contact between the Si_3_N_4_/Si_3_N_4_ sliding pairs because of the formation of a lubricating protective film. In addition, since FLG is easily torn under high Hertz contact pressure, disordered FLG with a much smaller particle size is formed [22]. FLG then wraps the abrasions to prevent direct contact between the sliding pairs. The reduction of the COF and anti-wear properties may be related to these mechanisms. A lubricating protective film was also confirmed in this paper, as measured by EDS analysis and Raman spectroscopy. The lubricating protective film effectively avoids direct contact between sliding pairs, thereby reducing the wear of the surface and decreasing the average friction coefficient.

## 4. Conclusions

The effects of FLG as a potential lubricating additive have been studied in detail in different experimental conditions. The results clearly show that FLG has a good effect on Mdol in improving the tribological properties of Si_3_N_4_/Si_3_N_4_ sliding pairs for high speeds and low loads, as well as decreasing temperature rise for low speeds regardless of the load level. The FLG content in Mdol has little effect on the width of the disks in the same conditions. On the contrary, it can reduce the WMD of the Si_3_N_4_ disks. This paper presented a lubrication mechanism to explain the test results. The presence of lubricating protective film can effectively avoid the direct contact between the sliding pairs. In addition, abrasive particles are coated with FLG. As a result, it reduced damage to the surface of the wear scar. FLG as an additive in lubricating oil can provide a good cooling effect for the Si_3_N_4_/Si_3_N_4_ sliding pairs, which can be attributed to the high thermal conductivity of FLG.

## Figures and Tables

**Figure 1 micromachines-11-00160-f001:**
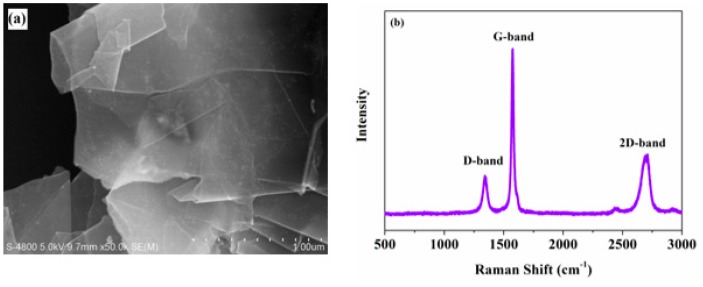
Structure images of few-layer graphene (FLG). (**a**) Scanning electron microscope (SEM) image; (**b**) Raman spectrum.

**Figure 2 micromachines-11-00160-f002:**
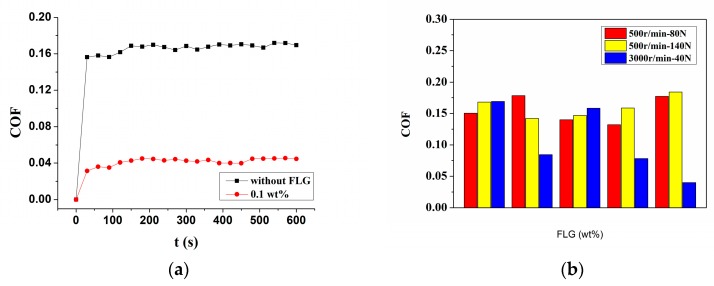
Friction coefficient (COF) results for Si_3_N_4_/Si_3_N_4_ sliding pairs under three working conditions. (**a**) COF with a 40 N load and 3000 r·min^−1^ rotating speed with and without 0.1 wt% FLG; (**b**) comparison of COF results under three different working conditions.

**Figure 3 micromachines-11-00160-f003:**
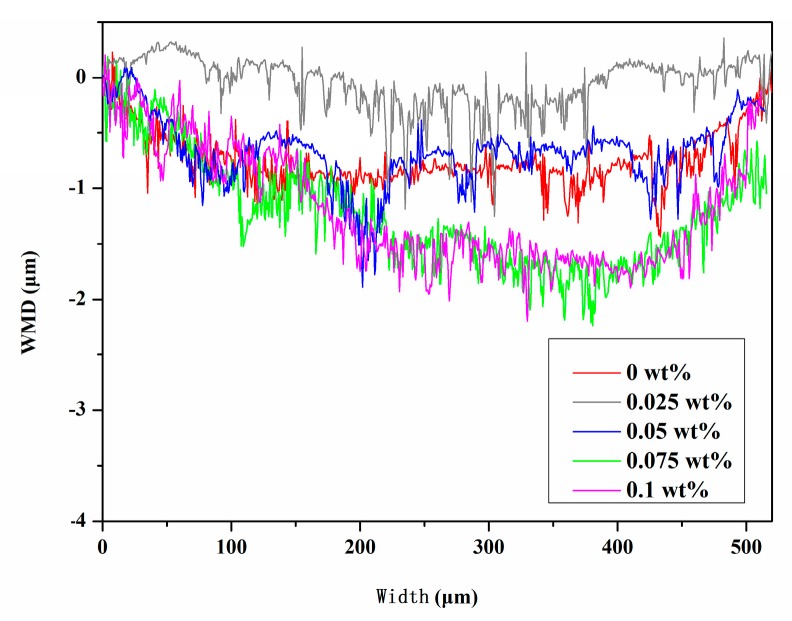
Wear mark depth (WMD) profile curve of the Si_3_N_4_ disk lubricated by different FLG contents.

**Figure 4 micromachines-11-00160-f004:**
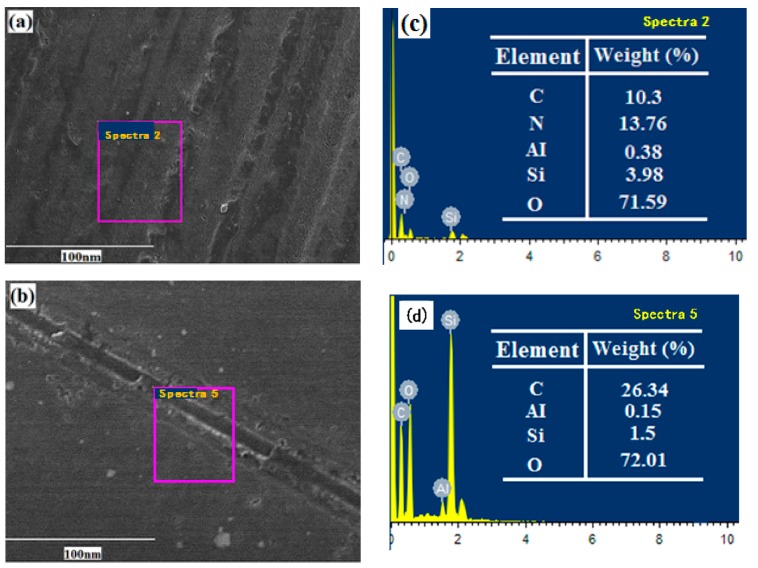
SEM image of worn surfaces lubricated by (**a**) pure Mdol and (**b**), Mdol containing 0.1 wt% FLG; (**c**,**d**) Energy-dispersive X-ray spectroscope (EDS) maps of the areas pointed out in (**a**,**b**).

**Figure 5 micromachines-11-00160-f005:**
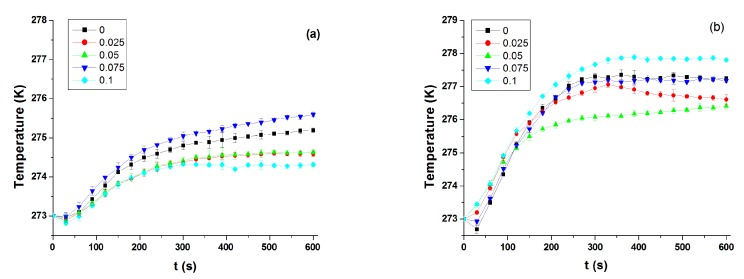
Temperature rise for Si_3_N_4_/Si_3_N_4_ sliding pairs under different conditions. (**a**) 80 N load and 500 r·min^−1^ rotational speed, (**b**) 140 N load and 500 r·min^−1^ rotational speed, (**c**) 40 N load and 3000 r·min^−1^ rotational speed, and (**d**) comparison of temperature rise results of the above.

**Figure 6 micromachines-11-00160-f006:**
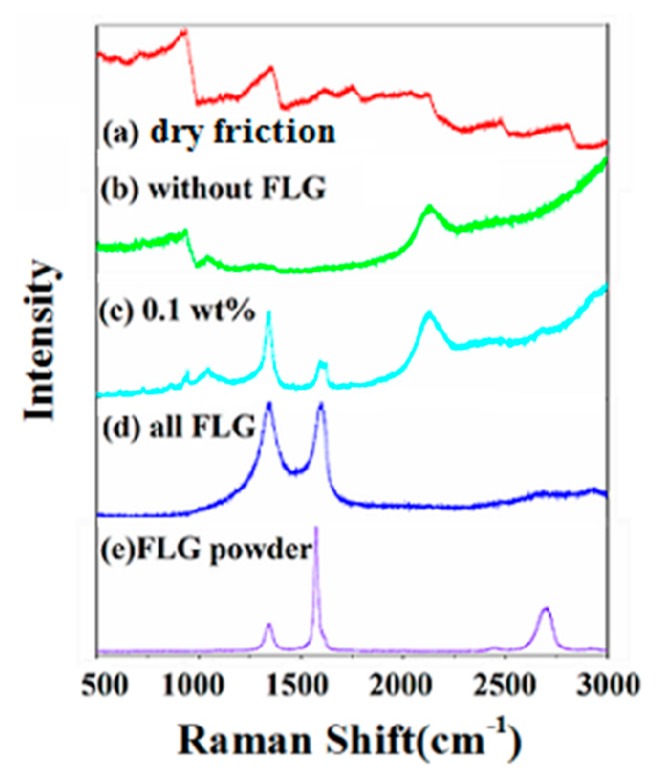
Raman spectra of the rubbing surfaces of the Si_3_N_4_ disks. (**a**) Without any lubrication; (**b**–**d**) lubricated by Mdol containing different contents of FLG; (**e**) FLG powder.

**Figure 7 micromachines-11-00160-f007:**
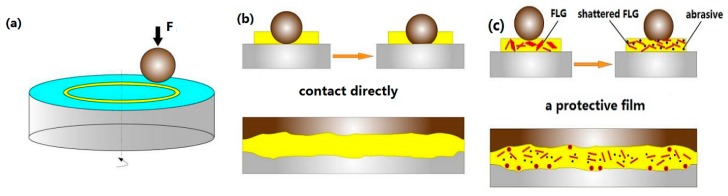
Schematic mechanism for FLG in Mdol. (**a**) Schematic diagram of the friction experiment, (**b**) lubrication diagram of pure lubricating oil, and (**c**) lubrication diagram of FLG suspended in base oil.

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
