# Peer review of "Experimental Study on the Lubrication and Cooling Effect of Graphene in Base Oil for Si3N4/Si3N4 Sliding Pairs"

_micromachines, 2020, doi:10.3390/mi11020160_

Round 1

Reviewer 1 Report

There are some concerns about this article.

The physical reasons provided by the authors to explain the addition of graphene on cooling is mostly speculation without sound proofs. The lubricant heat is mostly generated by the viscous dissipation which is determined by the sliding velocity, film thickness and viscosity. The heat convection and heat conduction normally play the secondary roles that the former part carries the heat downstream and the latter part deliver the heat away from the lubricant film. The authors may consult the article by Shyu(1998). With the addition of graphene, the particles could scrub the surfaces leading to additional heat depending on the particle size relative to the film thickness. It is hard to say how the high thermal conductivity affects the temperature rise relative to heat generation from the viscose dissipation as well as the friction heat by particles. Therefore, from my opinion, the authors need to provide more proofs to support their explanation.

-- Shyu, S.-H., 1998, Ranges of Validity for the Reynolds Equation and the Bulk-Flow Model in a Slider Bearing, Ph.D. Dissertation, The Pennsylvania State University,

Did the authors measure the film thickness? If not, how the authors justify the Fig. 7 and its discussion? The friction coefficient, wear depth, temperature and temperature rise should be noted using symbols, not acronym. The acronym like Flg etc. should be all capitalized. There are many English writing problems. For example, the paragraph from lines 181—184 is non- English; there are too many times that the authors wrote “…the Temperature…” instead of “…the temperature…”; the style for the figure captions are mostly wrong; there are many violations of space rule.

Reviewer 2 Report

Prior to the publication of this paper in Micromachines, the authors should respond to the following comments:

- What kind of supplier is China? This reference is used several times. Please be more specific.

- In the text is indicated that few-layer graphene "was obtained by physical methods and used directly without further purification". In a study where it is used in such low quantities, what reproducibility is obtained with this method of production?

- Similar problem with the oil used, Mobile DTE oil light is a commercial oil. In the text is indicated that "Mdol was a mineral oil without any additives". I don't think this is realistic in a commercial oil.

- In line 139 there is an error. A value of 80 N is indicated, but in figure caption, a value is only 40 N. It seems that the figure caption is correct.

- Do the authors have any explanation for the data shown in figure 3? Why is the war mark depth lower for lower concentrations of Flggraphene? How accurate is the method used to measure this wear mark depth? Especially since 0.1 % was the best value for the Afc.

- In my opinion, all Figure 4 should be on the same page. Apparently, the dimensions of the areas used to determine the surface composition appear different. Why?

- What's the meaning of Tr in Figure 5. How is calculated, at a fixed time or when the temperature is considered stable in each experiment? How many times this kind of experiment was performed? Please add, please add the uncertainty values in the measurements made and in the average values.

- Related with Figure 6. The band at 1439 cm-1 is indicated in the text as a "typical signal of Mdol". Have you performed Raman spectroscopy of pure Mdol? Can you assign that band to some kind of vibration?

- In figure 7, please add () to a, b and c, to better separate them from the text.

- Especially in the Results and Discussion section very low number of bibliographic references are used. Please, include a few more.

Minor comments:

Line 19 Please correct "manufracting"

Line 19 It seems that the verb "has" does not agree with the subject.

Line 35 Please consider change "as" to "with"

Line 36 Remove article "the complex"

Line 74 Please change the word "thesis" to "work". It seems more appropriate.

Line 85 Please add an article before “temperature”

Line 86 Please change "the temperature" to "a temperature"

Line 108 Please change "experiment" to "experiments".

Line 115 It appears that an article is missing before the word "physical".

Line 115 Please consider to change "particle" to "particles"

Line 117 The origin of the "Nanoplus-3-nanoparticle analyzer" is "America" or "USA"?

Line 154 Please change the article "an 40" to "a 40".

Line 163 Please add a space between Si3N4disks and use subscript for 3 and 4. check that all numbers for Si3N4 are in subscript.

Line 200 Please check the second "rise" in this line.

Line 207 Please, add letter (a) in the figure caption. Also in figure 6, it's shown Raman spectra for FLG powder as (e) but is not indicated in the figure caption.

Line 211 Please, change the sentence "This paper use this technique..." to "In this work, we use this technique.." or something similar.

Line 211 Please, add a space between Figure 8shows.

Line 217 Please, use for -1 superscript

Line 230 Please, change "to" to "in".

Line 242 Please, consider use "the" before "average"

Line 250 Please, consider delete "the" average...

Line 252 Please, consider use "a" good

Line 253 Please, consider use "the" high

Round 2

Reviewer 2 Report

In the first revision I asked to the authors to indicate the procedence of the different materials used, but "Si3N4 ceramic balls and disk were purchased from China". Please, correct.

There is a huge blank on page 5. I think figure 4 fits perfectly. Please check the correct paging of the document after this.

Author Response

Point 1: -In the first revision I asked to the authors to indicate the procedence of the different materials used, but "Si3N4 ceramic balls and disk were purchased from China". Please, correct.

Response 1: Done. The suppliers of Si3N4 ceramic balls and disk are specific in the manuscript.

Point 2: -There is a huge blank on page 5. I think figure 4 fits perfectly. Please check the correct paging of the document after this.

Response 2: Done. The blank is deleted. It is now well typesetting.